# Simulation Research on the Methods of Multi-Gene Region Association Analysis Based on a Functional Linear Model

**DOI:** 10.3390/genes13030455

**Published:** 2022-03-02

**Authors:** Shijing Li, Fujie Zhou, Jiayu Shen, Hui Zhang, Yongxian Wen

**Affiliations:** 1College of Computer and Information Science, Fujian Agriculture and Forestry University, Fuzhou 350002, China; 1191153009@fafu.edu.cn (S.L.); zhoufj.fz@139.com (F.Z.); 1201153016@fafu.edu.cn (J.S.); 1201153024@fafu.edu.cn (H.Z.); 2Institute of Statistics and Application, Fujian Agriculture and Forestry University, Fuzhou 350002, China

**Keywords:** functional linear model, multi-gene regions, association analysis, region weighted, loci weighted

## Abstract

Genome-wide association analysis is an important approach to identify genetic variants associated with complex traits. Complex traits are not only affected by single gene loci, but also by the interaction of multiple gene loci. Studies of association between gene regions and quantitative traits are of great significance in revealing the genetic mechanism of biological development. There have been a lot of studies on single-gene region association analysis, but the application of functional linear models in multi-gene region association analysis is still less. In this paper, a functional multi-gene region association analysis test method is proposed based on the functional linear model. From the three directions of common multi-gene region method, multi-gene region weighted method and multi-gene region loci weighted method, that test method is studied combined with computer simulation. The following conclusions are obtained through computer simulation: (a) The functional multi-gene region association analysis test method has higher power than the functional single gene region association analysis test method; (b) The functional multi-gene region weighted method performs better than the common functional multi-gene region method; (c) the functional multi-gene region loci weighted method is the best method for association analysis on three directions of the common multi-gene region method; (d) the performance of the Step method and Multi-gene region loci weighted Step for multi-gene regions is the best in general. Functional multi-gene region association analysis test method can theoretically provide a feasible method for the study of complex traits affected by multiple genes.

## 1. Introduction

Genetic analysis of rare variants is considered to be one of the most important components to compensate for the current deficiency of genetic variation, which has not yet been explained [1]. Although the lack of catalogs to speculate on the genotypes of rare variants and the high cost of sequencing technology have previously made it impossible to conduct very in-depth studies on rare variants [2,3], the development of high-throughput sequencing technology [4] has enabled scientists to obtain SNP data in a cheap and efficient way, as it contains a large amount of data on rare variants [5,6]. However, many previous tools and methods are designed for common variants, so there is still a lack of efficient and practical tools for rare variants association analysis. At present, single-marker association analysis is the most commonly used method of gene association analysis. However, if this method is directly applied to rare variants, it will be impossible to find loci with a moderate or low gene effect due to the limitations of single-marker association analysis [5,6]. The effect of a locus of a rare variant is small and not easily detected, and if the single-marker association analysis is used, many valuable association loci will be ignored.

To better find the weak effect sites, some methods to make the weak effect more significant by concentrating the association information of the whole gene region were proposed: a) the method based on fold (Collapsing Methods) [7,8,9]. By directly compressing multiple loci into a new variable, the associated rare variants with weak effects distributed at multiple loci are aggregated to make it easier to find; b) the method based on kern [10,11,12]. When the variance of a set of random variables is 0, the set of variables is made up of the same value, and therefore the kernel by method only needs to check whether the variance component of the group of effect estimates corresponding to all genotype variables in the whole region is 0; c) the method based on functional data analysis [13,14,15,16]. Functional data analysis converts the discrete loci into a continuous variable through the basis function, and only the coefficients of the effect estimation function corresponding to the continuous variable need to be tested in association analysis. There is also a strategy to consider and examine multiple loci at the same time and determine the significance of each locus [17,18], which is more effective than single-marker gene association analysis because it considers the interrelationships between multiple loci. The simplest method is to use the multivariate linear model as the test model for the multi-gene locus test [17]. However, when only a small part of the multiple gene loci included in the test are related, the large degree of freedom of the uncorrelated loci will lead to the loss of power.

The above-mentioned methods, whether based on gene region information aggregation or multi-gene locus analysis, have their advantages and disadvantages. Moreover, through continuous improvement and innovation of experts and scholars, the shortcomings of these methods have been constantly overcome and their performance has become more and more excellent. Since the single-gene region approach can aggregate small effects, the multi-locus approach can improve the analysis power by considering the interrelationship between multiple variables. If we combine these two methods, we can expect to obtain a multi-region analysis method with the advantages of both methods. In addition, the actual situation of phenotypic tend to be controlled by a few gene regions. Some of these gene region effects are apparent, some are weak, and a strong effect is easy recognize. However, a weak effect can easily be concealed by a stronger effect, even considering that this part of the phenotypic is controlled by the effect of apparent genetic regions.

At present, some scholars have carried out research on the combination of the two ideas, i.e., the aggregation of genetic information in gene regions and the use of interrelationship among multi-gene regions: One of them is Turkmen and Lin [19], who further extended the statistical test PDT (Pedigree Disequilibrium Test [20]) and FBAT (Family-Based Association Test [21]), and proposed Block analysis methods. Firstly, the specific approach is to divide the gene sequence to be analyzed into a block-by-block in a certain way and assume that the variants within the region are interdependent, but that the relationship between the blocks is mutually independent; secondly, PDT or FBAT methods are used to analyze the loci of each block; thirdly, the results in the region are generalized by means of the squares sum of standardized variances. After the statistics of each small block are obtained, the squares sum of the statistics is calculated again. After the aggregation of the information twice, a statistic subject to chi-square distribution is obtained, which is used as the gene association analysis statistics of the large gene region composed of these blocks. This method assumes that the loci inside the block are interdependent and the information inside the block is aggregated by PDT and FBAT methods, assuming that the blocks are independent of each other. The other method is by Ayers and Cordell [22], who improved the two methods of Group Lasso and Group Sparse Lasso [23], enabling them to distinguish common and rare variants within a group. This method can test multiple groups at the same time, in which one group is treated as a variable and the relationship between different groups is considered.

The analysis method of a single gene region based on functional data is a method to express the high-density genetic markers as functional data through integration and analyze the region through a linear model. Many experts have shown that this is an effective way to improve the power of gene association [13,14,24]. If the functional linear model considers multiple gene regions at the same time, it can not only improve the power by considering the interaction between multi-gene regions, but also isolate the effects of each gene region through the characteristics of the linear model, making the gene regions with weak effects more obvious and easier to find. Therefore, we will explore the method of gene association analysis of multi-gene regions based on functional data analysis, hoping to find a method with higher power and better detection of association loci with weak effects, and provide some references for researchers interested in this field in the future.

## 2. Materials and Methods

### 2.1. Statistic Model

#### 2.1.1. Common Multi-Gene Region Method

Suppose that there are n individuals in a genetic population. The genome region [0,T] is constructed by SNP sequences $t_{1}\leq t_{2}\cdots \leq t_{M}$ for genetic association analysis under the group structure and is not included. Let $y_{i}$ be the quantitative trait value of *i*-th individual and the population structure of the sample is not considered, so the traditional linear genetic model can be expressed as
(1)$$y_{i}=\mu_{0}+\sum_{j=1}^{M}x_{ij}\beta_{j}+\varepsilon_{i}\;i=1,2,\cdots ,n.$$
where $\mu_{0}$ is the overall mean of the model, $x_{ij}$ is a genotype profile (if A and a represent a pair of alleles, then when the genotype of *i*-th individual is AA, $x_{ij}$ is taken as 2; when the genotype of *i*-th individual is Aa, it is taken as 1; when the genotype of *i*-th individual is aa, is 0). $\beta_{j}$ represents the effect coefficient of genetic marker, $\varepsilon_{i}\sim N(0,\;\sigma^{2})$, $\sigma^{2}$ is the environmental genetic variance, M is the number of genetic markers. With the increase of the number of genetic markers, the freedom degree of the model gradually increases, and the multicollinearity among variables becomes more and more serious, eventually leading to the reduction of estimation accuracy and power. This is especially true when the genetic markers are low-frequency variations. When the discrete variants are at ultrahigh density, the discrete variants in an interval are as continuous, and the functional linear model (FLM) can be used instead of the multiple linear genetic model:(2)$$y_{i}=\mu_{0}+\int_{0}^{T}X_{i}(t)\beta (t)dt+\varepsilon_{i}\;i=1,2,\cdots ,n.$$
where $\varepsilon_{i}$ is an independent and normal distribution with zero mean and variance $\sigma^{2}$, and [0,T] represents the genomic region under consideration. The discrete genetic markers $x_{ij}$ in Equation (1) are converted into continuous genetic markers function $X_{i}(t)$ in Equation (2). At this time $X_{i}(t)$ is a random function, and the effects of genetic markers $\beta_{j}$ are also converted into a continuous genetic effect function β(t).

##### Step

The functional linear genetic model of single-gene region is generally in the following form
(3)$$y_{i}=\mu_{0}+\int_{0}^{T}X_{i}(t)\beta (t)dt+\varepsilon_{i},\;i=1,2,3,\cdots ,n.$$

When the single-gene region is extended to the multi-gene region, the original linear genetic model becomes the following form
(4)$$y_{i}=\mu_{0}+\sum_{p=1}^{P}\int_{0}^{T}X_{pi}(t)\beta_{p}(t)dt+\varepsilon_{i},\;i=1,2,3,\cdots ,n.$$

Suppose that there are *P* gene regions. There are SNP sequences $t_{1}^{p}<t_{2}^{p}<\cdots <t_{M}^{p}$ for *p*-th (p=1,2,⋯,P) gene region. Every gene region is [0,T]. For every region, the lower bound of the interval is converted to zero, and the upper bound of the interval is converted to T. According to the method of functional data analysis, a set of basis functions $\varphi_{p1}(t),\varphi_{p2}(t),\cdots \varphi_{pK_{G}}(t)$ and the coefficient $d_{pi1},d_{pi2},\cdots ,d_{piK_{G}}$ can be used to expand $X_{pi}(t)$ as
(5)$$X_{pi}(t)=\sum_{k=1}^{K_{G}}d_{pik}\varphi_{pk}(t),\;p=1,2,\cdots ,P;\;i=1,2,\cdots ,n.$$

Similarly, a set of basis functions $\phi_{p1}(t),\phi_{p2}(t),\cdots ,\phi_{pK_{\beta }}(t)$ and coefficient $b_{p1},b_{p2},\cdots ,b_{pK_{\beta }}$ can be used to expand $\beta_{p}(t)$ as
(6)$$\beta_{p}(t)=\sum_{k^{\prime }=1}^{K_{\beta }}b_{pk^{\prime }}\phi_{pk^{\prime }}(t),\;p=1,2,\cdots ,P.$$
the expansion of $X_{pi}(t)$ and $\beta_{p}(t)$, the model (4) becomes
(7)$$y_{i}=\mu_{0}+\sum_{p=1}^{P}\sum_{k=1}^{K_{G}}\sum_{k^{\prime }=1}^{K_{\beta }}d_{pik}\int_{0}^{T}\varphi_{pk}(t)\phi_{pk^{\prime }}(t)dt\cdot b_{pk^{\prime }}+\varepsilon_{i}\begin{array}{cc} , & i=1,2,3,\cdots ,n \end{array}.$$

Let $d_{pi}={\left[d_{pi1},d_{pi2},\cdots ,d_{piK_{G}}\right]}_{K_{G}\times 1}^{T}$, $b_{p}={\left[b_{p1},b_{p2},b_{p3},\cdots b_{pK_{\beta }}\right]}_{K_{\beta }\times 1}^{T}$, as well as
$$\Phi_{p}={\left[\begin{array}{ccc} \int_{0}^{T}\varphi_{p1}(t)\phi_{p1}(t)dt & \cdots & \int_{0}^{T}\varphi_{p1}(t)\phi_{pK_{\beta }}(t)dt\\ \vdots & \cdots & \vdots \\ \int_{0}^{T}\varphi_{pK_{G}}(t)\phi_{p1}(t)dt & \cdots & \int_{0}^{T}\varphi_{pK_{G}}(t)\phi_{pK_{\beta }}(t)dt \end{array}\right]}_{K_{G}\times K_{\beta }}$$

Model (7) becomes
(8)$$y_{i}=\mu_{0}+\sum_{p=1}^{P}d_{pi}^{T}\Phi_{p}b_{p}+\varepsilon_{i}\begin{array}{cc} , & i=1,2,3,\cdots ,n \end{array}.$$
where $\varphi_{p1}(t),\varphi_{p2}(t),\cdots \varphi_{pK_{G}}(t)$ and $\phi_{p1}(t),\phi_{p2}(t),\cdots ,\phi_{pK_{\beta }}(t)$ are a set of orthonormal basis. Usually, we choose the same basis function for $\varphi_{p1}(t),\varphi_{p2}(t),\cdots \varphi_{pK_{G}}(t)$ and $\phi_{p1}(t),\phi_{p2}(t),\cdots ,\phi_{pK_{\beta }}(t)$. Therefore, the above genetic model can be further simplified as
(9)$$y_{i}=\mu_{0}+\sum_{p=1}^{P}d_{pi}^{T}b_{p}+\varepsilon_{i}\begin{array}{cc} , & i=1,2,3,\cdots ,n \end{array}.$$

Model (8) becomes model (9). At this point, the genetic model is transformed into the ordinary multiple linear regression model of Equation (9), for which variables can be screened by stepwise regression [25,26,27]. Because only the interrelationship between whole gene regions and traits is discussed, $d_{pi}$ represents the genetic information of *p*-th gene region, so $d_{pi}$, which represents the whole information of a gene region, is considered to be added to the model as a “variable”. p gene regions should be screened as p variables.

There are three ways of screening variables for regression: forward selection, backward selection, forward selection, and backward selection. Here, the backward selection method is performed, where all gene regions to be analyzed are put into the model at the beginning, and then some gene regions are removed step-by-step until a reduced model is obtained. AIC (Akaike Information Criterion) information criteria will be used as the basis for each step to determine which gene regions need to be removed from the model
(10)AIC=nln(Rss/n)+2•K,
where Rss represents the squares sum of the residuals for the current model, and K represents the number of unknown variables, that is, the sum of the number of elements of all $d_{pi}$. After deciding which $d_{pi}$ to remove, we hypothetically delete each $d_{pi}$ existing in the current model and calculate the AIC, which is made up of the rest of the gene region. We find the model corresponding to the minimum AIC and then proceed to the next step. By repeating the above steps, until deleting any of the gene regions in the model does not make the AIC of the model smaller, we now have a very reduced model. Finally, the partial F test commonly used in the multiple linear regression model was used to test each $d_{pi}$ in the model, and the corresponding *p*-value was calculated as the evaluation basis for the association between gene regions and quantitative traits.

##### Multi-SLoS

Lin et al. [28] proposed a locally sparse functional linear model (SLoS method, Smooth and Locally Sparse method). By adding fSCAD (Functional Smoothly Clipped Absolute Deviation) penalty function on the basis of smoothing penalty term, the functional linear model has the ability to identify the sparse part of the estimated effect value $\hat{\beta }(t)$ and compress the estimated value to null. In the paper, a single region is taken as an example, but a model for multi-regions was also proposed:(11)$$y_{i}=\mu_{0}+\sum_{p=1}^{P}\int_{0}^{T}X_{pi}(t)\beta_{p}(t)dt+\varepsilon_{i}\begin{array}{cc} , & i=1,2,3,\cdots ,n \end{array}.$$

According to the description of the paper, to estimate the corresponding $\beta (t)={\left(\beta_{1}(t),\beta_{2}(t),\cdots \beta_{P}(t)\right)}^{T}$, we only need to solve the corresponding loss function:(12)$$\begin{array}{ll} Q(\beta ,\mu_{0})= & \frac{1}{n}\sum_{i=1}^{n}{\left[y_{i}-\mu_{0}-\sum_{p=1}^{P}\int_{0}^{T}X_{pi}(t)\beta_{p}(t)dt\right]}^{2}\\ & +\sum_{p=1}^{P}\gamma_{p}{\left\|D^{m}\beta_{p}\right\|}^{2}+\sum_{p=1}^{P}\frac{M}{T}\int_{0}^{T}p_{\lambda_{k}}^{\prime }(\left|\beta_{p}(t)\right|)dt \end{array}$$

Specific algorithms can be found in Lin et al. [28]. In order to distinguish the SLoS method for a single region, we refer to it as the Multi-SLoS method. The Multi-SLoS method has the ability of local sparse, that is, it can identify null and no-null in β(t). We hope to use the local sparsity ability of this model for gene association test, which involves the problem of the model test. That is, the significance test problem of individual gene regions in the model (2) is presented based on the Multi-SLoS method. The following is a detailed description of how to conduct the test based on the work of Lin et al. [28] for multi-gene region association analysis.

The functional linear model of a single gene region can be transformed into a multivariate linear model of several variables (the number of basis functions plus intercept), and then directly test whether the estimated coefficients of the basis functions are all null [13,29,30]. Multi-regions association analysis can be done similarly. However, when multi-gene regions are tested, not only must more variables be tested, but also the degrees of freedom of the model should be adjusted. In addition, we found that the results obtained after adjusting the degrees of freedom of the Multi-SLoS method as follows would be more consistent with the features of the method and the actual results in the follow-up study of polygenic regions. The reasons for the adjustment of degrees of freedom are given below.

Let the null gene regions denote gene regions where estimated effect values ${\hat{\beta }}_{p}(t)(p=1,2,\cdots ,P)$ are all null, and non-null gene regions denote gene regions where estimated effect value ${\hat{\beta }}_{p}(t)$ are not all null. The Multi-SLoS method will directly compress the estimated effect values ${\hat{\beta }}_{p}(t)$ of null gene regions to null and identify the non-null and null gene regions from *p* gene regions, regardless of whether the Multi-SLoS method is correct in distinguishing the non-null and null gene regions (the results can be seen in the later simulation). The regions where ${\hat{\beta }}_{p}(t)$ is compressed to null have no effect on the estimated results, which means that these regions have been directly identified as ${\hat{\beta }}_{p}(t)=0$ at a certain estimation stage of the model and have no effect on the estimated model. Then, the degrees of freedom of these gene regions should be removed from the calculation of the model. Similarly, there are some sub-regions where the effect is also compressed to null in non-null gene regions, which means that these sub-regions also have no effect on the estimated model, and the degrees of freedom of these sub-regions should be deducted.

Combined with the adjustment of degrees of freedom, the partial F test of the Multi-SLoS method is given below.

Let B represent the set of subscripts of gene regions with non-null effect, and b represent one element of the set B. For the above linear genetic model, the following method is used to test the association between gene region b and quantitative traits.


(a)Calculate the sum of square residuals of the full model including all non-null gene regions:(13)$${\hat{y}}_{i}=\mu_{0}+\sum_{k\in A}\int_{0}^{T}X_{ki}(t)\beta_{k}(t)dt\begin{array}{cc} , & i=1,2,3,\cdots ,n \end{array}.$$
(14)$$SSE(full)={\sum_{i=1}^{n}\left(y_{i}-{\hat{y}}_{i}\right)}^{2}$$(b)Calculate the sum of square residuals of the reduced model excluding gene region a:(15)$${\hat{y}}_{i}=\mu_{0}+\sum_{k\in B-b}\int_{0}^{T}X_{ki}(t)\beta_{k}(t)dt$$
(16)$$SSE(reduced)={\sum_{i=1}^{n}\left(y_{i}-{\hat{y}}_{i}\right)}^{2}$$(c)Adjustable degrees of freedom:


The adjusted freedom degree of SSE (full) (freedom_adj_(full)) should be: the number of individuals (n)—the sum of the number of non-null basis function coefficients in all non-null gene regions—1.

The adjusted freedom degree of SSE (reduced)–SSE (full) (freedom_adj_(reduced)) should be: the number of non-null basis function coefficients in gene region a.


(d)Calculate the corresponding values of *F* and *p* value
$$F=\frac{\frac{SSE(reduced)-SSE(full)}{freedom_{adj}(reduced)}}{\frac{SSE(full)}{freedom_{adj}(full)}}\sim F(freedom_{adj}(reduced),freedom_{adj}(full)$$


#### 2.1.2. Multi-Gene Region Weighted Method

In common gene association analysis, common variants can be easily identified if the associated loci contain both rare and common variants, but rare variants are difficult to detect because of their micro effects. For the association analysis of multi-gene regions, a similar situation is likely to occur—only the associated regions with rare variants are difficult to identify if the associated loci exist in the regions only with common variants and the regions only with rare variants at the same time. The common solution to this problem in gene-association analysis is to assign different weights to different types of variants. The same approach is used to assign different weights to different gene regions, by assigning weights to different types of gene regions to eliminate differences due to different allele frequencies rather than different degrees of association with phenotypic values.

##### Weighted SLoS (W-SLoS)

The SLoS approach has been applied to polygenic regions in Section 2.1.1, which asks the question: would it further improve the power if different weights were given to different types of gene regions? The loss function of the SLoS method is:(17)$$\begin{array}{ll} Q(\beta ,\mu_{0})= & \frac{1}{n}\sum_{i=1}^{n}{\left[y_{i}-\mu_{0}-\sum_{p=1}^{P}\int_{0}^{T}X_{pi}(t)\beta_{p}(t)dt\right]}^{2}\\ & +\sum_{p=1}^{P}\gamma_{p}{\left\|D^{m}\beta_{p}\right\|}^{2}+\sum_{p=1}^{P}\frac{M}{T}\int_{0}^{T}p_{\lambda_{p}}^{\prime }(\left|\beta_{p}(t)\right|)dt \end{array}$$

It can be seen from the loss function that different gene regions can be assigned different weights by adjusting parameters $\gamma_{p}$ and $p_{\lambda_{p}}^{\prime }$. Therefore, based on the research of Lin et al. [28], we have made appropriate adjustments to the code in the *slos* package of R language provided by Lin et al., so that the method is not only theoretically feasible but also runs smoothly in the actual program. Finally, the Weighted SLoS method only increases the weight compared to the Multi-SLoS method, and the same statistical test can be used to test the significance of each gene region.

#### 2.1.3. Multi-Gene Region Loci Weighted Method

Although it is possible to distinguish rare variants from common variants and then divide them into rare variants regions and common variants regions for analysis in the multi-gene region analysis, it is more common in the actual situation that both common variants and rare variants exist in a gene region to be analyzed. Therefore, the multi-gene region loci weighted method is proposed, which is a more general method of combining functional data analysis by assigning different weights to each locus within each region rather than to the gene region, as in Section 2.1.2.

##### Multi-Gene Region Loci Weighted Step (LW-Step)

Similar to Section 2.1.1, there are *n* individuals in a genetic population. The genome region [0,T] is constructed by SNPs sequences $t_{1}<t_{2}<\cdots <t_{M}$ for genetic association analysis under no group structure. Accordingly, the genetic markers are $x_{i1},x_{i2},\cdots ,x_{iM}(i=1,2,\cdots ,n)$. Let $y_{i}$ be the quantitative trait value of the *i*-th individual and the population structure of the sample is not considered, and the traditional linear genetic model can be expressed as
(18)$$y_{i}=\mu_{0}+\sum_{j=1}^{M}x_{ij}\beta_{j}+\varepsilon_{i},\;i=1,2,\cdots ,n.$$

With the increase of the number of genetic markers, the functional linear model (FLM) can be used instead of the multiple linear genetic model
(19)$$y_{i}=\mu_{0}+\int_{0}^{T}X_{i}(t)\beta (t)dt+\varepsilon_{i},\;i=1,2,\cdots ,n.$$

A set of basis functions $\varphi_{1}(t),\varphi_{2}(t),\varphi_{3}(t),\cdots ,\varphi_{K}{}_{{}_{G}}(t)$ and coefficients $d_{i1},d_{i2},d_{i3},\cdots d_{iK_{G}}$ can be used to expand $X_{i}(t)$ as
(20)$$X_{i}(t)=\sum_{k=1}^{K_{G}}d_{ik}\varphi_{k}(t)$$

According to functional analysis method [31], let $X_{i}=[x_{i1},x_{i2},\cdots ,x_{iM}]$ represent the gene data vector for the *i*-th individual, then,
$$d_{i}={\left[d_{i1},d_{i2},\cdots ,d_{iK_{G}}\right]}^{T},\;\varphi (t)={\left[\varphi_{1}(t),\varphi_{2}(t),\cdots ,\varphi_{K_{G}}(t)\right]}^{T},\;\varphi ={\left[\varphi (t_{1}),\varphi (t_{2}),\cdots ,\varphi (t_{M})\right]}^{T},$$

And
$$\varphi =\left[\begin{array}{c} \varphi^{T}(t_{1})\\ \varphi^{T}(t_{2})\\ \vdots \\ \varphi^{T}(t_{M}) \end{array}\right]={\left[\begin{array}{cccc} \varphi_{1}(t_{1}) & \varphi_{2}(t_{1}) & \cdots & \varphi_{K_{G}}(t_{1})\\ \varphi_{1}(t_{2}) & \varphi_{2}(t_{2}) & \cdots & \varphi_{K_{G}}(t_{2})\\ \vdots & \vdots & \cdots & \vdots \\ \varphi_{1}(t_{M}) & \varphi_{2}(t_{M}) & \cdots & \varphi_{K_{G}}(t_{M}) \end{array}\right]}_{M\times K_{G}}$$

Then $X_{i}(t)=d_{i}^{T}\varphi (t),i=1,2,\cdots ,n.$ According to the functional data analysis methods, there are
(21)$$\varphi d_{i}=\left[\begin{array}{c} d_{i}^{T}\varphi (t_{1})\\ d_{i}^{T}\varphi (t_{2})\\ \vdots \\ d_{i}^{T}\varphi (t_{M}) \end{array}\right]=\left[\begin{array}{c} \sum_{k=1}^{K_{G}}\varphi_{k}(t_{1})d_{ik}\\ \sum_{k=1}^{K_{G}}\varphi_{k}(t_{2})d_{ik}\\ \vdots \\ \sum_{k=1}^{K_{G}}\varphi_{k}(t_{M})d_{ik} \end{array}\right]=\left[\begin{array}{c} X_{i}(t_{1})\\ X_{i}(t_{2})\\ \vdots \\ X_{i}(t_{M}) \end{array}\right]=X_{i}(t)$$

The coefficient $d_{i}$ is solved in a smooth way
(22)$$PENSSE_{\lambda_{x}}(x)={\left(X_{i}^{T}-\varphi d_{i}\right)}^{T}(X_{i}^{T}-\varphi d_{i})+\lambda_{x}\int_{0}^{T}{\left[D^{2}X(t)\right]}^{2}dt,$$
(23)$$\int_{0}^{T}{\left[D^{2}X(t)\right]}^{2}dt=\int_{0}^{T}d_{i}^{T}[D^{2}\varphi (t)]{\left[D^{2}\varphi (t)\right]}^{T}d_{i}dt=d_{i}^{T}R_{2}d_{i}.$$

Here, $R_{2}$ is a penalty matrix,
(24)$${\left[R_{2}\right]}_{jk}=\int_{0}^{T}[D^{2}\varphi_{j}(t)][D^{2}\varphi_{k}(t)]dt,\;j=1,2,\cdots ,K_{G};k=1,2,\cdots ,K_{G}.$$

The solution result is ${\hat{d}}_{i}={\left[\varphi^{T}\varphi +\lambda_{x}R_{2}\right]}^{-1}\varphi^{T}X_{i}^{T}$, then
(25)$$X_{i}(t)=X_{i}\varphi {\left[\varphi^{T}\varphi +\lambda_{x}R_{2}\right]}^{-1}\varphi (t),i=1,2,\cdots ,n.$$

In addition,
(26)$$\beta (t)=\sum_{k^{\prime }=1}^{K_{\beta }}b_{k^{\prime }}\phi_{k^{\prime }}(t)={\left[\phi (t)\right]}^{T}b,$$
where $\phi (t)={\left[\phi_{1}(t),\phi_{2}(t),\cdots ,\phi_{K_{\beta }}(t)\right]}^{T},$ $b={\left(b_{1},b_{2},\cdots ,b_{K_{\beta }}\right)}^{T},$ combined $X_{i}(t),$
β(t) and functional linear model
(27)$$y_{i}=\mu_{0}+\int_{0}^{T}X_{i}(t)\beta (t)dt+\varepsilon_{i},$$
the following can be obtained
(28)$$y_{i}=\mu_{0}+X_{i}Wb+\varepsilon_{i},\;i=1,2,\cdots ,n.$$
where $W=\varphi {\left[\varphi^{T}\varphi +\lambda_{x}R_{2}\right]}^{-1}\int_{0}^{T}\varphi (t)[\phi (t){]}^{T}dt.$

Next, as in Belonogova et al. [32], a M×M diagonal matrix Θ was designed, where each element on the diagonal of the matrix corresponds to the weight of genotype data $X_{i}=[x_{i1},x_{i2},\cdots ,x_{iM}]$. The weight can be determined by Beta distribution
$$Beta(MAF_{ij},a_{1},a_{2}),\;i=1,2,\cdots ,n,j=1,2,\cdots ,M.$$
where $a_{1},a_{2}$ are the preset parameters, and $MAF_{ij}$ represents the *j*-th genotype frequency of the *i*-th individual. The diagonal matrix Θ is embedded to the simplified functional linear equation [32]
(29)$$y_{i}=\mu_{0}+X_{i}\Theta Wb+\varepsilon_{i},i=1,2,\cdots ,n.$$

This gives us a weighted functional linear function. This is the method of a single gene region, corresponding to the functional linear equation of multiple gene regions, which can be
(30)$$y_{i}=\mu_{0}+\sum_{p=1}^{P}\int_{0}^{T}X_{pi}(t)\beta_{p}(t)dt+\varepsilon_{i},\;i=1,2,\cdots ,n.$$

All the assumptions about gene regions are the same before. According to the specific situation of each region, add a weight matrix $\Theta_{p}$ to assign different weights to the loci in the region, then
(31)$$y_{i}=\mu_{0}+\sum_{p=1}^{P}X_{pi}\Theta_{p}W_{p}b_{p}+\varepsilon_{i},\;i=1,2,\cdots ,n.$$

The above statement is used to better explain the method in theory, in fact, the actual processing is not so complicated as in theory. Returning to the fitting of genotype data, the problem of loci weighting can be viewed from another perspective. First, let $X_{i}^{*}=X_{i}\Theta$, and then expand $X_{i}^{*}$ with the same functional smoothing parameters, so that
(32)$$X_{i}(t)=X_{i}\varphi {\left[\varphi^{T}\varphi +\lambda_{x}R_{2}\right]}^{-1}\varphi (t)$$
(33)$$X_{i}^{*}(t)=X_{i}^{*}\varphi {\left[\varphi^{T}\varphi +\lambda_{x}R_{2}\right]}^{-1}\varphi (t)=X_{i}\Theta \varphi {\left[\varphi^{T}\varphi +\lambda_{x}R_{2}\right]}^{-1}\varphi (t)$$

These are the same smooth parameters, basis functions, number of basis functions, and nodes. The value of $\varphi {\left[\varphi^{T}\varphi +\lambda_{x}R_{2}\right]}^{-1}\varphi (t)$ is decided by the above factors. $\varphi {\left[\varphi^{T}\varphi +\lambda_{x}R_{2}\right]}^{-1}\varphi (t)$ is the same as the expansion of $X_{i}(t)$ and $X_{i}^{*}(t)$. The difference between $X_{i}(t)$ and $X_{i}^{*}(t)$ is the weight matrix Θ. This result can be used to deduce the single gene loci weighted functional linear model as follows
(34)$$\begin{array}{ll} y_{i} & =\mu_{0}+X_{i}\Theta Wb+\varepsilon_{i}\\ & =\mu_{0}+X_{i}\Theta \varphi {\left[\varphi^{T}\varphi +\lambda_{x}R_{2}\right]}^{-1}\int_{0}^{T}\varphi (t){\left[\phi (t)\right]}^{T}dtb+\varepsilon_{i}\\ & =\mu_{0}+X_{i}^{*}\varphi {\left[\varphi^{T}\varphi +\lambda_{x}R_{2}\right]}^{-1}\int_{0}^{T}\varphi (t){\left[\phi (t)\right]}^{T}dtb+\varepsilon_{i}\\ & =\mu_{0}+\int_{0}^{T}X_{i}^{*}\varphi {\left[\varphi^{T}\varphi +\lambda_{x}R_{2}\right]}^{-1}\varphi (t){\left[\phi (t)\right]}^{T}bdt+\varepsilon_{i}\\ & =\mu_{0}+\int_{0}^{T}X_{i}^{*}(t)\beta (t)dt+\varepsilon_{i} \end{array}$$
where $W=\varphi {\left[\varphi^{T}\varphi +\lambda_{x}R_{2}\right]}^{-1}\int_{0}^{T}\varphi (t)[\phi (t){]}^{T}dt,$ i=1,2,⋯,n.

It can be seen from the derived results that the single gene loci weighted linear functional linear model can be understood as the weighted transformation of the original genotype data into new functional data $X_{i}^{*}(t)$, and then the functional linear model can be established by using $X_{i}^{*}(t)$. We extend this result into the multi-gene region weighted functional linear model, and all the assumptions are the same as the above section. The model becomes the ordinary multi-gene region functional linear model
(35)$$y_{i}=\mu_{0}+\sum_{p=1}^{P}\int_{0}^{T}X_{pi}^{*}(t)\beta_{p}(t)dt+\varepsilon_{i},\;i=1,2,\cdots ,n$$

Then,
(36)$$\begin{array}{l} y_{i}=\mu_{0}+\sum_{p=1}^{P}\int_{0}^{T}X_{pi}^{*}(t)\beta_{p}(t)dt+\varepsilon_{i}\\ \;=\mu_{0}+\sum_{p=1}^{P}\int_{0}^{T}X_{pi}^{*}\varphi_{p}^{}{\left[\varphi_{p}^{T}\varphi_{p}+\lambda_{x}R_{2}\right]}^{-1}\varphi_{p}(t){\left[\phi_{p}(t)\right]}^{T}b_{p}dt+\varepsilon_{i}\\ \;=\mu_{0}+\sum_{p=1}^{P}X_{pi}^{*}\varphi_{p}^{}{\left[\varphi_{p}^{T}\varphi_{p}+\lambda_{x}R_{2}\right]}^{-1}\int_{0}^{T}\varphi_{p}(t){\left[\phi_{p}(t)\right]}^{T}dtb_{p}+\varepsilon_{i} \end{array}$$
where $\varphi_{p}(t)={\left[\varphi_{p1}(t),\varphi_{p2}(t),\cdots ,\varphi_{pK_{G}}(t)\right]}^{T},$
$$\phi_{p}(t)={\left[\phi_{p1}(t),\phi_{p2}(t),\cdots ,\phi_{pK_{\beta }}(t)\right]}^{T},\;\varphi_{p}={\left[\varphi_{p}(t_{1}^{p}),\varphi_{p}(t_{2}^{p}),\cdots ,\varphi_{p}(t_{M}^{p})\right]}^{T},$$

And
$$\varphi_{p}=\left[\begin{array}{c} \varphi_{p}^{T}(t_{1}^{p})\\ \varphi_{p}^{T}(t_{2}^{p})\\ \vdots \\ \varphi_{p}^{T}(t_{M}^{p}) \end{array}\right]={\left[\begin{array}{cccc} \varphi_{p1}(t_{1}^{p}) & \varphi_{p2}(t_{1}^{p}) & \cdots & \varphi_{pK_{G}}(t_{1}^{p})\\ \varphi_{p1}(t_{2}^{p}) & \varphi_{p2}(t_{2}^{p}) & \cdots & \varphi_{pK_{G}}(t_{2}^{p})\\ \vdots & \vdots & \cdots & \vdots \\ \varphi_{p1}(t_{M}^{p}) & \varphi_{p2}(t_{M}^{p}) & \cdots & \varphi_{pK_{G}}(t_{M}^{p}) \end{array}\right]}_{M\times K_{G}},p=1,2,\cdots ,P.$$

Let $d_{pi}^{*}=X_{pi}^{*}\varphi_{p}{\left[\varphi_{p}^{T}\varphi_{p}+\lambda_{x}R_{2}\right]}^{-1}$
$$\Phi_{p}={\left[\begin{array}{ccc} \int_{0}^{T}\varphi_{p1}(t)\phi_{p1}(t)dt & \cdots & \int_{0}^{T}\varphi_{p1}(t)\phi_{pK_{\beta }}(t)dt\\ \vdots & \cdots & \vdots \\ \int_{0}^{T}\varphi_{pK_{G}}(t)\phi_{p1}(t)dt & \cdots & \int_{0}^{T}\varphi_{pK_{G}}(t)\phi_{pK_{\beta }}(t)dt \end{array}\right]}_{K_{G}\times K_{\beta }}$$

Then
(37)$$y_{i}=\mu_{0}+\sum_{p=1}^{P}d_{pi}^{*}\Phi_{p}b_{p}+\varepsilon_{i},\;i=1,2,\cdots ,n.$$

For the same reasons as in Step method, when $\varphi_{p1}(t),\varphi_{p2}(t),\cdots \varphi_{pK_{G}}(t)$ and $\phi_{p1}(t),\phi_{p2}(t),\cdots ,\phi_{pK_{\beta }}(t)$ are a set of orthonormal basis and the same basis function, the difference between different individual data is mainly reflected in the coefficients. Therefore, it is reasonable to simply take the coefficients of the basis function as a new variable as the basis of subsequent method operations, and simplify the model as follows
(38)$$y_{i}=\mu_{0}+\sum_{p=1}^{P}d_{pi}^{*}b_{p}+\varepsilon_{i},\;i=1,2,\cdots ,n.$$

The model becomes a multivariate linear model with the coefficients of the basis functions in the region as the new variables. The difference between the two methods compared with the Step method is that, at this time, the coefficients are obtained by functional data analysis of genotype data after weighting. The simplified model treats each gene region as a ’variable’ for stepwise regression and conducts a partial F test on the final simplified model to obtain the significance level of each gene region. The method is called the Multi-gene region loci weighted Step (LW-Step).

##### Multi-Gene Region Loci Weighted SLoS Method (LW-SLoS)

For the multi-gene region, the model becomes the ordinary multi-gene region functional linear model
(39)$$y_{i}=\mu_{0}+\sum_{p=1}^{P}\int_{0}^{T}X_{pi}^{*}(t)\beta_{p}(t)dt+\varepsilon_{i},\;i=1,2,\cdots ,n.$$

The SLoS method can be used to solve the model, and the statistical test method proposed in Multi-SLoS method can be used to test the model. Finally, in order to distinguish between other types of SLoS methods, we will call this method LW-SLoS, which means Loci Weight SLoS.

### 2.2. Design of Simulation Evaluation

#### 2.2.1. Simulation of Common Multi-Gene Region Method

In the simulation analysis of the common multi-gene region method, 25 gene regions are generated at a time and spliced together as the "multi-gene region" to be analyzed at one time. We design four different multi-gene regions: (a) rare variant multi-gene regions, i.e., all variants within the regions are rare variants (MAF (Minor Allele Frequency) < 0.01); (b) common variant multi-gene regions, i.e., all variants within the regions are common variants; (c) the hybrid variant multi-gene regions (15 gene regions are common variant gene regions and 10 gene regions are rare variant gene regions); (d) the hybrid variant multi-gene regions: each gene region is a mixture of 60% rare variants and 40% common variants. The purpose of designing these multi-gene regions is to better find out the specific scenarios applicable to multi-gene analysis and the different manifestations of multi-gene region analysis in different scenarios.

The rare variant gene regions and common variant gene regions in the multi-gene region are generated as follows: for rare variant gene regions, each time, a 5 kb gene segment is randomly selected as a gene region from the rare haploid dataset (the R language SKAT package [33] contains a set of such data produced by simulating allele frequency and linkage imbalance information in European populations) with a length of 200 kb, then 2000 individuals are randomly selected from 10,000 individuals twice to synthesize the diploid region of the rare variant genes. For common variant gene regions, firstly, the allele frequencies of gene loci are generated uniformly distributed; secondly, according to the frequency of each locus, haploid datasets of common variants with the same structure and size as the rare variants dataset are generated by simple random sampling; finally, the common variant gene regions are generated in the same way as the rare variant gene regions. Among the 4 multi-genic regions: the rare variant multi-gene regions are directly composed of 25 rare variant gene regions; the common variant multi-gene regions are directly composed of 25 common variant gene regions; the hybrid variant multi-gene regions are composed of 15 common variant gene regions and 10 rare variant gene regions, in addition, the splicing order of the two gene regions is random; the hybrid variant multi-gene regions is generated in this way—firstly, make 25 regions of rare variants and 25 regions of common variants (rare variant regions and common variant regions must be of the same size and structure); secondly, 40% of the rare variants are randomly selected from the first rare variant gene region and filled with genotypes corresponding to first common variant gene region. This is repeated until the generation of the 25-th gene region is completed.

In power simulation, the associated loci should be assumed as the target of the analysis and the quantitative traits should be simulated as the analysis objects. Therefore, in each simulation, five of the 25 gene regions splicing “multi-gene regions” are selected as the associated gene regions, and then three loci are randomly selected from each associated gene region as the associated loci. The generation of simulated traits adopted the additive effect model. At the same time, three different scenarios are made for the effect of the associated loci: Scenario I, all effect directions are positive; Scenario II, the effect direction of all associated loci are negative in two of the five associated regions; Scenario III, the effect direction of one locus in each associated region is negative. The absolute value of effect value is determined by the following effect model
(40)$$|\beta (t_{j})|=|\log_{10}(2•MAF_{t_{j}})|/4\times 1.5,\;t_{1}\leq \cdots \leq t_{j}\cdots \leq t_{M}.$$
where $MAF_{t_{j}}$ represents the minimum allele frequency of the *t_j_*-th genotype as the associated locus in the associated gene regions. In the false-positive proportional simulation, random numbers were generated through normal distribution N(0,0.1) as the phenotypic values of the simulation, since it was assumed that no associated gene loci existed in the gene region.

The multi-gene regions that are composed of 25 gene regions are analyzed for every simulation. Each gene region is simulated 100 times under different association effect hypotheses. There are 2500 (100×25) gene regions analyzed. In each scenario, 5 gene regions are assumed to be associated regions, and 20 gene regions are assumed to be unrelated regions. That is, there are 500 associated gene regions and 2000 unrelated gene regions for every case. Under given significance level α based on every model, every method and every scenario, the number of significant gene regions is $n_{1}$. The number of significant gene regions, but no significant gene regions, in fact, is $n_{2}$. The power is $\frac{n_{1}}{500}$ and the false positive rates (Type I error rates) are $\frac{n_{2}}{2000}$.

In order to compare the Step and Multi-SLoS method of multi-gene region analysis with the single-gene region functional method, the SLoS method and FLM method are also performed in this simulation. The single-gene region method analyzes the sub-regions of the simulated multi-gene region one by one and then summarizes the results to test the multi-gene regions. The FLM method is proposed by Svishcheva et al. [14] as a functional gene region analysis method for family and population gene data. The author provides a package, “FREGAT”, written in the R language, that contains the computer program for the method. Moreover, the method can also be used for genetic association analysis in populations with no family relationship. For SLoS method and Multi-SLoS method, the compression parameter and smoothing parameter are set as 0.1 and 0.1 in the common variants multi-gene regions simulation; as 0.01 and 0.01 in the rare variants multi-gene regions simulation; and as 0.05 and 0.05 for the hybrid variants multi-gene regions simulation and the hybrid variants multi-gene regions simulation. The fitting of SLoS and Multi-SLoS models requires the "SLoS" package of R language [28], but the statistical part needs to be supplemented by the R language code written by us. In the process of simulation, genotype data needs to be converted into functional data, all gene regions are smoothed by 25 Fourier basis functions, the number of nodes is equal to the number of variants in the gene regions, and the distance of nodes is equidistant. The number of basis functions of the effect function uses the default settings for the appropriate software. The basis functions and node settings are the same in subsequent computational simulations, except for additional instructions.

#### 2.2.2. Simulation of Multi-Gene Region Weighted Method

We designed three different multi-gene regions: (a) rare variants multi-gene regions, i.e., all variants within the regions are rare variants (MAF < 0.01); (b) common variants multi-gene regions, i.e., all variants within the regions are common variants; (c) the hybrid variants multi-gene regions: 15 are common variants gene regions and 10 are rare variants gene regions. Here, a simulation of the multi-gene region weighted method is only aimed at hybrid variants multi-gene regions analyzed in the simulation of common multi-gene region method. Besides, there are two common variant gene regions and three rare variant gene regions for five associated regions. In addition to counting the power and false positive rates of common and rare variant gene sub-regions in multi-genic regions, the rest of the settings are similar with those simulations of the common multi-gene region method, in order to count the power and false positive rates of the common variants gene regions and the rare variants gene regions in the multi-gene regions respectively. We illustrate the rare variants gene regions as an example: the power of rare variants gene sub-regions is the total number of significant and true associated gene regions in the rare variants gene sub-regions divided by the total number of rare variants gene sub-regions in the multi-gene regions. False positive rates of rare variants gene sub-regions: the total number of significant but unrelated gene regions in rare variants gene sub-regions is divided by the total number of rare variants gene sub-regions in multi-gene regions.

Three methods are used in this simulation: Step, FLM, and Weighted SLoS (W-SLoS). During the simulation of the W-SLoS method, the smoothing parameters of common variants gene sub-regions are 0.02 and the compression parameters are 0.05. The smoothing parameter and compression parameter of rare variants gene sub-regions are 0.01 and 0.0025, respectively. All settings for the nodes and the effect of genetic variations are similar as the computer simulations for the common multi-gene region method (Section 2.2.1).

#### 2.2.3. Simulation of Multi-Gene Region Loci Weighted Method

The multi-gene regions generated by the simulation in the multi-gene regions loci weighted analysis is composed of 10 mixed variants gene regions. The proportion of rare variants in the 10 regions of mixed variants are as follows: [0.7,0.8,0.6,0.95,0.9,0.95,0.7,0.9,0.8,0.6], respectively. Common variants and rare variants of every sub-region are random distributions for simulated multi-gene regions. The gene loci are weighted in the sub-regions. The rare variants and common variants in each gene region are generated in the same way as the common multi-gene regions simulations. This multi-gene region will be analyzed 100 times during the simulations. During the power simulation, the associated gene regions are preset as 2-th, 4-th, 5-th, 7-th, and 10-th gene regions, the proportion of rare variants in the corresponding gene regions is [0.8,0.95,0.9,0.7,0.6], and the 4-th and 5-th gene regions are adjacent. Although the associated gene regions are determined, the associated loci in each gene region are randomly extracted from loci with a minimum allele frequency of less than 0.02 within the corresponding gene regions in each simulation. In addition, the model and method used to simulate phenotypic values and the effect value scenario settings of the associated loci are the same as those in the common multi-gene regions simulations. Suppose that there are three scenarios for the effect of the associated loci: Scenario I, all effect directions of all associated loci are positive for gene loci; Scenario II, the effect directions of all associated loci are negative for the 4-th and 7-th gene region; Scenario Ⅲ, choose a locus at random form associated loci and the effect direction of that locus is negative in each associated region.

For power and false positive rates in the simulation, it is assumed that in the set of regions that are significant under a certain condition, the number of regions that are truly correlated and significant is $n_{1}$, while the number of uncorrelated regions and identified as significant is $n_{2}$, and then the total power in the simulation is: $\frac{n_{1}}{500}$, and the total false positive rates (Type I error rates) are: $\frac{n_{2}}{500}$. Suppose that the number of regions identified as significant in the 2-th, 4-th, 5-th, 7-th, 10-th gene regions is $n_{i1},(i=2,4,5,7,10)$, respectively. Then, the powers of the corresponding gene sub-regions are $\frac{n_{i1}}{100},(i=2,4,5,7,10)$. Suppose that the number of regions in the 1-th, 3-th, 6-th, 8-th, 9-th gene regions identified as significant is: $n_{i2},(i=1,3,6,8,9)$ (in fact, those gene regions are no significant), then the false positive rates of corresponding gene sub-regions are $\frac{n_{i2}}{100}$,(i=1,3,6,8,9).

Four methods are used in the simulation: Step, Loci Weighted Step (LW-Step), Multi-SLoS, and Loci Weighted SLoS (LW-SLoS). The smoothing and compression parameters of the Multi-SLoS and LW-Step methods are set to 0.001. The multi-gene region loci weighted method needs to set the weights of different loci. In this paper, it is realized by setting the weight matrix. Here, the weight matrix is set as the diagonal matrix, and the weight of the corresponding gene loci of the diagonal elements on the matrix is generated through the beta distribution:$$Beta(MAF_{i},1,10).$$

## 3. Results

### 3.1. Results of the Common Multi-Gene Region Method

It can be seen from Table 1 that the power of the Step method is the highest among the four methods at each significance level in the rare variants multi-gene regions, and the power of the Multi-SLoS method is not higher than that of FLM and Step, but it is also slightly higher than that of SLoS; in hybrid variants multi-gene regions Ⅰ and hybrid variants multi-gene regions Ⅱ the power of the Step method is higher than the FLM method, but the power of the Multi-SLoS method is not higher than the SLoS method. Therefore, in terms of power performance, multi-gene region analysis has a comparative advantage in rare variants multi-gene regions, and the power of the Step method is the best in four simulated gene regions. By comparing the power on different types of gene regions, the SLoS and Multi-SLoS method have higher power for multi-gene regions with common variants; the Step method makes it less powerful in multi-gene regions that contain common variants; the FLM method is more effective in multi-gene regions consisting of only rare or common variants, but the performance of hybrid variants multi-gene regions I is very strange.

Combined with the simulation results of false positive rates in Table 2, the results are further analyzed. In rare variants multi-gene regions, the Multi-SLoS method has a higher false positive rate than the SLoS method, and the Step method has a higher false positive rate than the FLM method under different effect directions and higher significance level. The false positive rates in both the Step method and Multi-SLoS method are very low in the common variants multi-gene regions. In the hybrid variants multi-gene regions II, the Step method and Multi-SLoS method compared to the FLM and SLoS methods have lower false positive rates. Therefore, the addition of common variants to the sub-regions of the multi-gene regions has the effect of reducing the false positive rates.

On the one hand, the multi-gene region Step method based on functional data analysis has the best power and performance in false positive rates. Indeed, it can better find some associated regions that cannot be found in single gene regions, especially some associated gene regions with micro effects. On the other hand, the Multi-SLoS method has no significant advantages over the SLoS method and needs further improvement and adjustment.

### 3.2. Results of Multi-Gene Region Weighted Method

As can be seen from the power simulation results in Table 1 and Table 3, the highest power of the Multi-SLoS method is 76.2% and of the SLoS method is 78.4%. The lowest power of W-SLoS is 84% for the hybrid variant multi-gene region I (see Table 1). Therefore, the SLoS method with weighted multi-gene regions has a significant improvement in power. The FLM method still does not perform well in this case. Combined with the previous simulation of the FLM method, it has a good performance when the gene sub-region in the multi-gene regions is of the same type, but the performance is not good when the multi-gene regions are mixed with multiple types of gene regions. It can be known that this method may be more suitable for the detection of multi-gene regions with the same type of gene sub-regions. The Step method is still the most powerful of the four methods, both in terms of overall power and in gene regions with common or rare variants. Almost all of the common variant gene regions are identified, and rare variants gene regions have lower power than that of the common variants gene regions. In general, the three methods can more easily identify rare variants gene regions, but common variants gene regions still require higher power.

Table 4 shows the simulation results of false positive rates. It is obvious that the false positive rates of FLM are larger than that of other methods. Among three methods, the false positive rates of the W-SLoS method are the lowest, followed by the Step method. In general, the performance of W-SLoS is better than that of Step on false positive rates from Table 4. It means the W-SLoS method is not as effective as the Step method for detecting gene regions, but it is more reliable for gene regions selected by the W-SLoS method.

By weighting the sub-regions, the performance of the multi-gene region SLoS method exceeds that of the single-gene region SLoS method. This indicates that weighting can further improve the power of the polygenic region analysis model.

### 3.3. Results of Multi-Gene Region Loci Weighted Method

Table 5 and Table 6 show the power and the false positive rates simulation results of the multi-gene region loci weighted method. In the power simulation results, there are three unweighted methods: FLM, Multi-SLoS and Step, and the highest power of these methods is 94.2%. The highest power of the two loci-weighted methods is 98.2%. This suggests that multi-gene regions loci weighted methods can indeed improve the power in this simulation. For false positive rates simulation, the false positive rates of LW-Step are higher than that of Step, but lower than that of the other three methods. The LW-SLoS method has much higher false positive rates than the LW-Step method, but the false positive rates of LW-SLoS approach that of the LW-Step method as the significance level increased gradually.

Since the multi-gene regions in each simulation are fixed, so are the associated gene sub-regions as well, and the power and false positive rates of each gene sub-region are calculated. Figure 1 and Figure 2, respectively, show the power of the 2-th, 4-th, 5-th, 7-th, and 10-th associated gene regions, and the false positive rates of the 1-th, 3-th, 6-th, 8-th, and 9-th unassociated gene regions. Figure 1 shows that the power of different gene regions is different under different effect hypotheses, the proportion of common and rare variants in a gene region affects power of the region. Figure 2 shows that in general, the LW-SLoS method has higher false positive rates in the 1-th and 3-th gene regions. The proportions of rare variants in these two gene regions are 0.7 and 0.6, which are the second and the smallest in the five gene regions.

In general, the multi-gene region loci weighted method has an advantage in the multi-gene regions, where the proportions of various variants in each gene region are different. Moreover, the simulation results of power and false positive rates show that such results are not simply lowering the threshold, but it improves the power of the analysis at reasonable false positive rates. The simulation further analyses the power and false positive rates of different gene sub-regions in the same multi-gene regions. The results show that the presence of some common variants in the sub-gene regions could improve the power of the method and reduce false positive rates.

## 4. Discussion

In this paper, a total of five analysis methods are proposed for the analysis of multi-gene regions, which can be divided into three categories: the common multi-gene region method, the multi-gene region weighted method, and the multi-gene region loci weighted method. the Step method merged the two ideas together with the gene information of the region and the relationship among the gene regions. The simulation results showed that the power of the Step method is higher than that of the FLM and SLoS method for a single gene region, even for that of the region-weighted SLoS method. It means the power is improved for considering the relationship among gene regions. The multi-gene region loci weighted method is the most complex but also the most effective. Its power simulation results are much higher than the unweighted single gene region analysis method and the false positive ratio is much lower than the single gene region analysis method. For the SLoS method, the simulation results of the common multi-gene region method are only slightly better than the common single-gene region analysis method, which also shows that even the simplest multi-gene region analysis method can effectively improve the power of the analysis compared with the single-gene region analysis method. In general, the simulation results of the Step method and LW-Step method are better methods for the associated analysis of multi-gene region. By modified freedom degree of test statistic F, the multi-SLoS, W-SLoS and LW-SLoS are feasible for the associated analysis of multi-gene region. Compared with the rare variant multi-gene region, the associated analysis result of the common variant multi-gene region is better than using the multi-gene region analysis method.

The multi-gene region loci weighted method is a further expansion and extension of the weighting idea of Belonogova et al. [32]. However, there are some differences between our work and Belonogova’s paper: firstly, the weighted idea was not applied to a multi-gene region in his paper; secondly, the coefficient of functional data is estimated by the smooth method in our paper and the least square method in Belonogova’s paper.

The Fourier basis function is selected to fit the genotype data when the functional expansion is carried out. The reason as to why we chose the Fourier basis function is that some studies have compared the Fourier basis to the spline basis, achieving similar results (the previously cited papers on functional gene-association analysis have compared it in their papers). However, some people question how genes can be represented by periodic Fourier bases. Perhaps they both extracted the same amount of information for the gene regions they wanted to summarize in their way, and after a lot of people compared the results of the two bases, there was only a very small difference. Even though the Fourier basis function might look better in some cases, it is not good enough for the authors of those papers to conclude that the Fourier basis function is a better choice. Usually, both of the basis functions are used, so one can choose one of two types of basis functions. The reason why the Fourier basis is selected in this paper is that the Fourier basis only needs to determine the number of basis functions, while the spline basis not only needs to select the number of basis functions but also the order of the basis function, so the selection of the Fourier basis function is much simpler. It must choose the Fourier basis for the Step method and the LW-Step method.

Regarding the selection of compression parameters and smoothing parameters, the paper does not necessarily choose the weights that can give the method the best performance, but it basically selects the parameters according to the most common standards and methods (such as 10-fold cross-validation, etc.). The specific parameter selection strategy can be: Firstly, determine the selection criteria of parameters, then determine the approximate range of compression parameters and smoothing parameters according to the method and the actual situation, finally, the computer program is used to screen out the optimal parameters. This process must be repeated in the processing of actual data, because the genetic region composition of the actual data is constantly changing, and so it requires that the parameters change accordingly. However, in the simulation, the same situations in the genetic regions are made up the same way. Therefore, in order to save computing resources, a suitable parameter is directly selected in this paper.

One might ask: why not just analyze it as a larger gene region? Instead, we analyze it in the way of this multi-gene region, and the functional data can do this only by increasing the number of nodes and the number of basis functions. One reason is that the large gene regions can be divided into smaller gene regions, and then the multi-gene region method can be used to get more detailed test results. Another reason is that, as mentioned in the previous article, due to the consideration of the interrelationship between different gene regions, the multi-gene region method can better identify some regions with relatively weak effects and have higher power. So, compared to single-gene region testing, multi-gene region testing can detect gene regions where the effect is smaller or where the effect overlaps with that of other regions.

We focused on independent SNPs for common variants. Rare variants come from the rare haploid dataset (the R language SKAT package [33] contains a set of such data produced by simulating allele frequency and linkage imbalance information in European populations) with a length of 200 kb. However, we also simulate linkage disequilibrium in common variant multi-gene region of Scenario I based on the common multi-gene region method and multi-gene region weighted method. When the *r*^2^ measure of linkage disequilibrium is between 0.25 and 0.64, the power of the linkage disequilibrium based on the SLoS and Multi-SloS methods is higher than that of linkage equilibrium, but the false positive rates also increase significantly. The power of the linkage disequilibrium based on the Step method decreases slightly, and the false positive rates remain largely unchanged. The power and false positive rates of the linkage disequilibrium based on the FLM method does not change significantly. This indicates that the linkage disequilibrium among the gene loci causes SLoS and Multi-SLoS methods to more easily misidentify non-associated gene regions. The association analysis between gene region and quantitative trait is susceptible to linkage disequilibrium of gene loci for SLoS and Multi-SLoS methods. Then, the association analysis of gene regions for quantitative traits was unstable for SLoS and Multi-SLoS methods, while Step and FLM methods are more stable. The reason may be the parameter estimation of SLoS and Multi-SLoS method. Furthermore, we compare the simulation results of forward selection and backward selection and find that the power of the forward selection method is slightly higher than that of the back selection method, but the false positive rates of the forward selection method is far greater than that of the back selection method. Finally, we study the distribution of allele frequencies of gene loci. The results show that the power of following a normal distribution is higher than that of following a uniform distribution, which may be due to the fact that the MAF of the normal distribution is mostly smaller than that of the uniform distribution. According to the effect function expression, the effect size of the normal distribution is larger at this time and can make the locus easier to detect.

The model we considered is an ideal state, as it is only a study of basic assumptions, without considering group structure and population with relatives, and there is no missing genotype. In practice, it is inevitable that there will be missing genotypes. At this time, we can use statistical methods to fill in the missing data, and then convert the discrete genetic data into continuous functions. In future research, we will consider incorporating models of group structure and population with relatives.

## Figures and Tables

**Figure 1 genes-13-00455-f001:**
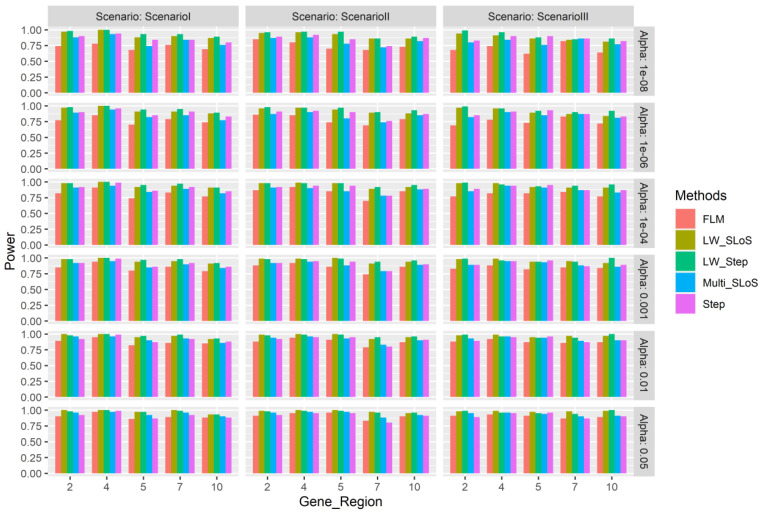
The simulation power of gene regions 2, 4, 5, 7, and 10 using the multi-gene region loci weighted method. Scenario I—all effect directions of all associated loci are positive for gene loci; Scenario II—the effect directions of all associated loci are negative for the 4-th and 7-th gene region; Scenario III—choose a gene locus at random and the effect direction of the gene loci is negative for every associated region.

**Figure 2 genes-13-00455-f002:**
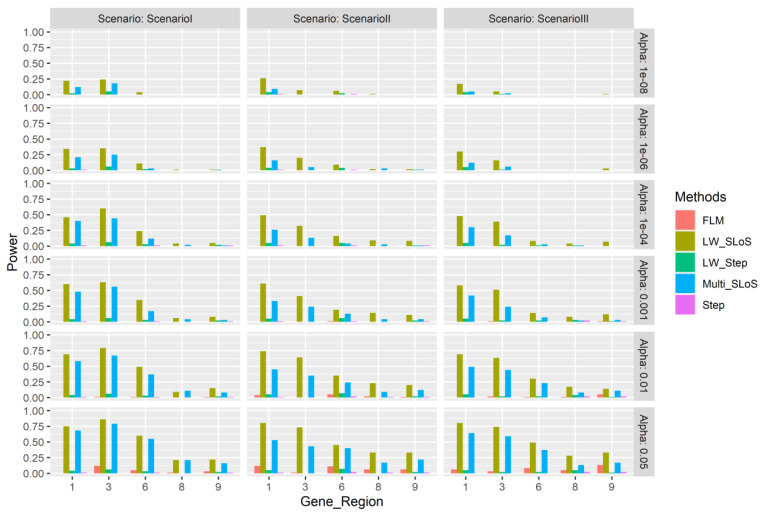
The simulation false positive rates of gene regions 1, 3, 6, 8, and 9 using the multi-gene region loci weighted method. Scenario I—all effect directions of all associated loci are positive for gene loci; Scenario II—the effect directions of all associated loci are negative for the 4-th and 8-th gene region; Scenario III—choose a gene locus at random and the effect direction of the gene loci is negative for every associated region.

**Table 1 genes-13-00455-t001:** Simulation results of the power for four types of multi-gene regions regarding the common multi-gene region method.

Gene Effect	α	Common Variant				Rare Variant				Hybrid Variant				Hybrid Variant			
		Multi-Gene Region				Multi-Gene Region				Multi-Gene Region I				Multi-Gene Region II			
		Step	Multi-SLoS	FLM	SLoS	Step	Multi-SLoS	FLM	SLoS	Step	Multi-SLoS	FLM	SLoS	Step	Multi-SLoS	FLM	SLoS
Scenario I	0.05	0.9560	0.8360	0.9680	0.8540	0.9880	0.7120	0.9820	0.7140	0.9720	0.7480	0.3120	0.7720	0.9000	0.8460	0.8980	0.8600
	0.01	0.9560	0.8300	0.9540	0.8500	0.9880	0.7020	0.9600	0.6880	0.9700	0.7440	0.2640	0.7700	0.8980	0.8460	0.8720	0.8520
	0.001	0.9480	0.8260	0.9240	0.8480	0.9880	0.6860	0.9200	0.6660	0.9640	0.7400	0.2420	0.7620	0.8900	0.8440	0.8620	0.8480
	1 × 10^−4^	0.9220	0.8120	0.8780	0.8360	0.9860	0.6820	0.8820	0.6580	0.9600	0.7320	0.2220	0.7540	0.8780	0.8240	0.8460	0.8260
	1 × 10^−6^	0.8600	0.7540	0.7920	0.7600	0.9780	0.6720	0.8380	0.6420	0.9200	0.7120	0.2000	0.7240	0.8660	0.8120	0.8020	0.8000
	1 × 10^−8^	0.7840	0.6600	0.6680	0.6500	0.9680	0.6660	0.7880	0.6300	0.8780	0.6900	0.1780	0.6800	0.8520	0.7920	0.7420	0.7560
Scenario II	0.05	0.9680	0.8560	0.9800	0.8860	0.9860	0.6900	0.9520	0.6900	0.9600	0.7340	0.3040	0.7520	0.8800	0.8240	0.8900	0.8540
	0.01	0.9660	0.8460	0.9640	0.8840	0.9860	0.6740	0.9400	0.6580	0.9600	0.7340	0.2640	0.7480	0.8800	0.8200	0.8560	0.8540
	0.001	0.9540	0.8360	0.9440	0.8840	0.9860	0.6600	0.9240	0.6480	0.9560	0.7320	0.2360	0.7400	0.8760	0.8160	0.8320	0.8320
	1 × 10^−4^	0.9360	0.8200	0.9080	0.8720	0.9840	0.6580	0.8980	0.6400	0.9480	0.7280	0.2280	0.7240	0.8720	0.8080	0.8100	0.8060
	1 × 10^−6^	0.8940	0.7440	0.8040	0.7920	0.9840	0.6460	0.8420	0.6280	0.9080	0.7040	0.2040	0.6880	0.8400	0.7820	0.7620	0.7620
	1 × 10^−8^	0.8080	0.6640	0.6900	0.6740	0.9680	0.6400	0.7980	0.6140	0.8560	0.6820	0.1860	0.6400	0.8120	0.7580	0.7180	0.7200
Scenario III	0.05	0.9560	0.8240	0.9740	0.8420	0.9940	0.6980	0.9660	0.6680	0.9480	0.7620	0.2880	0.7840	0.8900	0.8360	0.8900	0.8540
	0.01	0.9540	0.8200	0.9460	0.8420	0.9940	0.6660	0.9480	0.6300	0.9480	0.7580	0.2380	0.7760	0.8900	0.8340	0.8700	0.8500
	0.001	0.9380	0.8100	0.9120	0.8400	0.9940	0.6440	0.9200	0.6080	0.9380	0.7480	0.2180	0.7600	0.8880	0.8300	0.8440	0.8340
	1 × 10^−4^	0.9120	0.7940	0.8680	0.8280	0.9900	0.6280	0.8900	0.5940	0.9240	0.7320	0.2020	0.7380	0.8760	0.8160	0.8220	0.8140
	1 × 10^−6^	0.8420	0.6940	0.7680	0.7200	0.9860	0.6160	0.8340	0.5740	0.8860	0.7000	0.1840	0.6900	0.8540	0.7860	0.7780	0.7820
	1 × 10^−8^	0.7620	0.6000	0.6720	0.6340	0.9700	0.6000	0.7880	0.5540	0.8360	0.6500	0.1640	0.6180	0.8400	0.7580	0.7300	0.7340

Note: Scenario I—all effect directions are positive: Scenario II—the effect direction of all associated loci are negative in two of the five associated regions; Scenario III—the effect direction of one locus in each associated region is negative.

**Table 2 genes-13-00455-t002:** Simulation results of the false positive rate for four types of multi-gene regions regarding the common multi-gene region method.

Gene Effect	α	Common Variant				Rare Variant				Hybrid Variant				Hybrid Variant			
		Multi-Gene Region				Multi-Gene Region				Multi-Gene Region I				Multi-Gene Region II			
		Step	Multi-SLoS	FLM	SLoS	Step	Multi-SLoS	FLM	SLoS	Step	Multi-SLoS	FLM	SLoS	Step	Multi-SLoS	FLM	SLoS
Scenario I	0.05	0.0045	0.0000	0.0530	0.0000	0.0755	0.0685	0.0910	0.0310	0.0110	0.0515	0.2830	0.1055	0.0060	0.0060	0.0550	0.0290
	0.01	0.0045	0.0000	0.0085	0.0000	0.0750	0.0315	0.0410	0.0040	0.0110	0.0275	0.2315	0.0485	0.0060	0.0040	0.0135	0.0170
	0.001	0.0030	0.0000	0.0015	0.0000	0.0670	0.0200	0.0135	0.0000	0.0085	0.0085	0.2105	0.0050	0.0060	0.0015	0.0010	0.0035
	1 × 10^−4^	0.0005	0.0000	0.0000	0.0000	0.0580	0.0155	0.0075	0.0000	0.0055	0.0020	0.1975	0.0000	0.0050	0.0005	0.0000	0.0005
	1 × 10^−6^	0.0000	0.0000	0.0000	0.0000	0.0445	0.0095	0.0025	0.0000	0.0030	0.0000	0.1785	0.0000	0.0015	0.0005	0.0000	0.0000
	1 × 10^−8^	0.0000	0.0000	0.0000	0.0000	0.0380	0.0075	0.0005	0.0000	0.0025	0.0000	0.1595	0.0000	0.0005	0.0005	0.0000	0.0000
Scenario II	0.05	0.0025	0.0000	0.0430	0.0000	0.0615	0.0710	0.0920	0.0305	0.0070	0.0530	0.2645	0.1270	0.0045	0.0075	0.0455	0.0230
	0.01	0.0020	0.0000	0.0095	0.0000	0.0605	0.0475	0.0375	0.0045	0.0065	0.0325	0.2250	0.0560	0.0045	0.0050	0.0085	0.0135
	0.001	0.0005	0.0000	0.0005	0.0000	0.0540	0.0355	0.0095	0.0000	0.0035	0.0050	0.2000	0.0050	0.0020	0.0010	0.0005	0.0000
	1 × 10^−4^	0.0005	0.0000	0.0000	0.0000	0.0490	0.0295	0.0035	0.0000	0.0030	0.0020	0.1855	0.0005	0.0010	0.0000	0.0000	0.0000
	1 × 10^−6^	0.0000	0.0000	0.0000	0.0000	0.0385	0.0245	0.0000	0.0000	0.0020	0.0010	0.1675	0.0000	0.0005	0.0000	0.0000	0.0000
	1 × 10^−8^	0.0000	0.0000	0.0000	0.0000	0.0335	0.0155	0.0000	0.0000	0.0010	0.0010	0.1505	0.0000	0.0000	0.0000	0.0000	0.0000
Scenario III	0.05	0.0065	0.0000	0.0525	0.0005	0.0675	0.0875	0.1035	0.0365	0.0090	0.0565	0.2675	0.1220	0.0045	0.0050	0.0425	0.0280
	0.01	0.0060	0.0000	0.0110	0.0005	0.0665	0.0615	0.0455	0.0040	0.0080	0.0270	0.2255	0.0490	0.0040	0.0040	0.0065	0.0150
	0.001	0.0030	0.0000	0.0000	0.0000	0.0575	0.0530	0.0150	0.0000	0.0040	0.0095	0.2060	0.0085	0.0025	0.0010	0.0015	0.0015
	1 × 10^−4^	0.0015	0.0000	0.0000	0.0000	0.0510	0.0435	0.0065	0.0000	0.0030	0.0040	0.1955	0.0015	0.0015	0.0000	0.0005	0.0005
	1 × 10^−6^	0.0000	0.0000	0.0000	0.0000	0.0400	0.0325	0.0020	0.0000	0.0010	0.0025	0.1670	0.0000	0.0015	0.0000	0.0000	0.0000
	1 × 10^−8^	0.0000	0.0000	0.0000	0.0000	0.0320	0.0260	0.0005	0.0000	0.0000	0.0025	0.1470	0.0000	0.0005	0.0000	0.0000	0.0000

Note: Scenario I—all effect directions are positive: Scenario II—the effect direction of all associated loci are negative in two of the five associated regions; Scenario III—the effect direction of one locus in each associated region is negative.

**Table 3 genes-13-00455-t003:** Simulation results of the power for three types of multi-gene regions regarding the multi-gene region weighted method.

Gene Effect	α	Common Variant Multi-Gene Region			Rare Variant Multi-Gene Region			Hybrid Variant Multi-Gene Region I		
		W-SLoS	Step	FLM	W-SLoS	Step	FLM	W-SLoS	Step	FLM
Scenario I	0.05	0.9220	0.9600	0.9680	0.8900	0.9960	0.9700	0.9040	0.9680	0.2940
	0.01	0.9220	0.9600	0.9520	0.8900	0.9960	0.9480	0.9000	0.9680	0.2660
	0.001	0.9220	0.9480	0.9220	0.8900	0.9960	0.9320	0.8960	0.9580	0.2520
	1 × 10^−4^	0.9220	0.9260	0.8800	0.8900	0.9960	0.8940	0.8860	0.9540	0.2400
	1 × 10^−6^	0.8880	0.8580	0.7900	0.8900	0.9920	0.8360	0.8700	0.9280	0.2260
	1 × 10^−8^	0.8400	0.7800	0.6680	0.8860	0.9820	0.8060	0.8520	0.9020	0.2080
Scenario II	0.05	0.9360	0.9700	0.9800	0.8780	0.9940	0.9720	0.9260	0.9780	0.3040
	0.01	0.9360	0.9700	0.9580	0.8780	0.9940	0.9600	0.9260	0.9780	0.2680
	0.001	0.9360	0.9700	0.9280	0.8780	0.9940	0.9060	0.9220	0.9740	0.2460
	1 × 10^−4^	0.9360	0.9480	0.8940	0.8780	0.9920	0.8680	0.9160	0.9720	0.2360
	1 × 10^−6^	0.9120	0.8720	0.7940	0.8720	0.9900	0.8320	0.9040	0.9500	0.2240
	1 × 10^−8^	0.8740	0.7920	0.7080	0.8700	0.9880	0.7940	0.8880	0.9280	0.1840
Scenario III	0.05	0.9080	0.9520	0.9720	0.8920	0.9940	0.9680	0.8920	0.9680	0.3380
	0.01	0.9080	0.9520	0.9540	0.8920	0.9940	0.9340	0.8900	0.9680	0.2900
	0.001	0.9080	0.9420	0.8940	0.8920	0.9920	0.9000	0.8800	0.9640	0.2640
	1 × 10^−4^	0.9040	0.9060	0.8620	0.8920	0.9900	0.8800	0.8740	0.9540	0.2540
	1 × 10^−6^	0.8820	0.8440	0.7720	0.8900	0.9860	0.8280	0.8580	0.9320	0.2240
	1 × 10^−8^	0.8320	0.7700	0.6640	0.8840	0.9800	0.7940	0.8400	0.9040	0.1940

Note: Scenario I—all effect directions are positive: Scenario II—the effect direction of all associated loci are negative in two of the five associated regions; Scenario III—the effect direction of one locus in each associated region is negative.

**Table 4 genes-13-00455-t004:** Simulation results of the false positive rate for three types of multi-gene regions regarding the multi-gene region weighted method.

Gene Effect	α	Common Variant Multi-Gene Region			Rare Variant Multi-Gene Region			Hybrid Variant Multi-Gene Region I		
		W-SLoS	Step	FLM	W-SLoS	Step	FLM	W-SLoS	Step	FLM
Scenario I	0.05	0.0020	0.0065	0.0480	0.0070	0.0640	0.0960	0.0060	0.0110	0.2815
	0.01	0.0020	0.0055	0.0075	0.0065	0.0635	0.0415	0.0045	0.0105	0.2335
	0.001	0.0015	0.0010	0.0000	0.0060	0.0585	0.0140	0.0020	0.0065	0.2120
	1 × 10^−4^	0.0015	0.0000	0.0000	0.0060	0.0500	0.0060	0.0010	0.0040	0.1980
	1 × 10^−6^	0.0000	0.0000	0.0000	0.0045	0.0415	0.0015	0.0005	0.0025	0.1810
	1 × 10^−8^	0.0000	0.0000	0.0000	0.0035	0.0310	0.0000	0.0000	0.0015	0.1620
Scenario II	0.05	0.0015	0.0055	0.0520	0.0055	0.0615	0.0785	0.0150	0.0110	0.2670
	0.01	0.0015	0.0050	0.0090	0.0055	0.0610	0.0295	0.0085	0.0110	0.2300
	0.001	0.0015	0.0015	0.0010	0.0050	0.0570	0.0090	0.0040	0.0070	0.2120
	1 × 10^−4^	0.0005	0.0005	0.0000	0.0050	0.0515	0.0045	0.0020	0.0040	0.2015
	1 × 10^−6^	0.0000	0.0000	0.0000	0.0040	0.0440	0.0000	0.0000	0.0010	0.1870
	1 × 10^−8^	0.0000	0.0000	0.0000	0.0035	0.0355	0.0000	0.0000	0.0005	0.1715
Scenario III	0.05	0.0020	0.0070	0.0500	0.0070	0.0665	0.0820	0.0215	0.0145	0.2625
	0.01	0.0020	0.0065	0.0080	0.0060	0.0660	0.0290	0.0110	0.0140	0.2150
	0.001	0.0015	0.0035	0.0015	0.0050	0.0555	0.0090	0.0055	0.0090	0.1895
	1 × 10^−4^	0.0005	0.0005	0.0000	0.0050	0.0495	0.0025	0.0035	0.0065	0.1780
	1 × 10^−6^	0.0000	0.0000	0.0000	0.0045	0.0420	0.0000	0.0015	0.0035	0.1655
	1 × 10^−8^	0.0000	0.0000	0.0000	0.0035	0.0360	0.0000	0.0010	0.0025	0.1545

Note: Scenario I—all effect directions are positive: Scenario II—the effect direction of all associated loci are negative in two of the five associated regions; Scenario III—the effect direction of one locus in each associated region is negative.

**Table 5 genes-13-00455-t005:** Simulation results of the power for mixed variation region regarding the multi-gene region loci weighted method.

Gene Effect	α	LW-Step	LW-SLoS	Step	Multi-SLoS	FLM
Scenario I	0.05	0.9740	0.9800	0.9160	0.9420	0.9000
	0.01	0.9740	0.9680	0.9160	0.9220	0.8740
	0.001	0.9700	0.9560	0.9100	0.8920	0.8480
	1 × 10^−4^	0.9620	0.9500	0.9080	0.8800	0.8140
	1 × 10^−6^	0.9520	0.9340	0.8900	0.8540	0.7700
	1 × 10^−8^	0.9460	0.9240	0.8640	0.8300	0.7300
Scenario II	0.05	0.9760	0.9820	0.9060	0.9400	0.9100
	0.01	0.9740	0.9720	0.9060	0.9120	0.8780
	0.001	0.9700	0.9660	0.9000	0.8840	0.8520
	1 × 10^−4^	0.9620	0.9520	0.8940	0.8640	0.8380
	1 × 10^−6^	0.9500	0.9280	0.8720	0.8320	0.7860
	1 × 10^−8^	0.9300	0.9120	0.8540	0.8140	0.7520
Scenario III	0.05	0.9680	0.9820	0.9140	0.9320	0.9020
	0.01	0.9660	0.9720	0.9140	0.9240	0.8800
	0.001	0.9660	0.9560	0.9120	0.9020	0.8440
	1 × 10^−4^	0.9560	0.9400	0.9040	0.8800	0.8040
	1 × 10^−6^	0.9380	0.9060	0.8780	0.8500	0.7500
	1 × 10^−8^	0.9080	0.8720	0.8620	0.8060	0.7000

Note: Scenario I—all effect directions are positive: Scenario II—the effect direction of all associated loci are negative in two of the five associated regions; Scenario III—the effect direction of one locus in each associated region is negative.

**Table 6 genes-13-00455-t006:** Simulation results of the false positive rate for mixed variation region regarding the multi-gene region loci weighted method.

Gene Effect	α	LW-Step	LW-SLoS	Step	Multi-SLoS	FLM
Scenario I	0.05	0.0300	0.5280	0.0060	0.4780	0.0420
	0.01	0.0300	0.4420	0.0060	0.3620	0.0060
	0.001	0.0300	0.3440	0.0060	0.2560	0.0000
	1 × 10^−4^	0.0300	0.2780	0.0060	0.1980	0.0000
	1 × 10^−6^	0.0240	0.1640	0.0020	0.0980	0.0000
	1 × 10^−8^	0.0140	0.1000	0.0000	0.0600	0.0000
Scenario II	0.05	0.0280	0.5280	0.0100	0.3500	0.0720
	0.01	0.0280	0.4320	0.0100	0.2500	0.0240
	0.001	0.0260	0.2920	0.0060	0.1560	0.0020
	1 × 10^−4^	0.0220	0.2280	0.0060	0.0940	0.0000
	1 × 10^−6^	0.0180	0.1400	0.0040	0.0500	0.0000
	1 × 10^−8^	0.0120	0.0800	0.0040	0.0180	0.0000
Scenario III	0.05	0.0300	0.5280	0.0080	0.3800	0.0700
	0.01	0.0280	0.3860	0.0080	0.2700	0.0180
	0.001	0.0260	0.2860	0.0060	0.1560	0.0040
	1 × 10^−4^	0.0180	0.2120	0.0000	0.1020	0.0000
	1 × 10^−6^	0.0120	0.0980	0.0000	0.0360	0.0000
	1 × 10^−8^	0.0100	0.0460	0.0000	0.0140	0.0000

Note: Scenario I—all effect directions are positive: Scenario II—the effect direction of all associated loci are negative in two of the five associated regions; Scenario III—the effect direction of one locus in each associated region is negative.

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
