# Peer review of "Simulation Research on the Methods of Multi-Gene Region Association Analysis Based on a Functional Linear Model"

_genes, 2022, doi:10.3390/genes13030455_

Round 1

Reviewer 1 Report

My comments are in the uploaded word file.

Author Response

Our response is in the uploaded pdf file.

Reviewer 2 Report

This paper by Li et al. proposes a multi-gene region association analysis test method based on the functional linear model. Simulations were performed and a few conclusions regarding the power of different models in characterizing genetic variants associated with complex traits were made based on the simulating results. I think the topic of this work is of good importance and interest, but there are several points below needed to be further addressed.

Major comments:

  1. On page 3 line 132-135, why was the continuous representation named FLM? Can the authors explain the logic of this kind of naming? And why a continuous definition could reduce the degree of freedom of the model and the multicollinearity of a variable due to low-frequency variation?
  2. On page 4 line 145-147, I don’t see which terms describe gene-gene interaction? Is that inter-genetic effect ignored? It might need more detailed explanations.
  3. On page 4 line 165-167, the authors claim that the Phi_p does not provide information about genotypes, and later on it was left out of the equation. This part is confusing and more detailed explanations are needed.
  4. On page 5 line 180-196, this sounds like an optimization procedure based on the criteria of “local optimum” but perhaps does not guarantee the “global optimum”. I wonder what will the regression variables look like if the forward selection is used and how does that compare with the results from the backward selection used here.
  5. On page 5 line 204, Is this function adapted from Lin et al. (2017)? This looks the same as what the authors proposed in the current manuscript, or maybe it is the writing itself that leads to this confusion.
  6. On page 8 line 281-287, what would be the explicit mathematical formula look like in W-SLoS?

Minor comments:

  1. There are grammar flaws throughout the manuscript. For example, in line 121, “Let y_i is …” should be “Let y_i be …”. In line 132, it should be "degree of freedom". The authors should double check the texting.
  2. In Figure 1 for y-label, I suggest instead of starting from 0, start from some number like 0.5 to highlight the difference between different methods.

Author Response

(The authors gave the same response as above.)
